# When Language Models Fall in Love:
# Animacy Processing in Transformer Language Models

**Michael Hanna**
ILLC
University of Amsterdam
m.w.hanna@uva.nl

**Yonatan Belinkov**
Technion—IIT, Israel
belinkov@technion.ac.il

**Sandro Pezzelle**
ILLC
University of Amsterdam
s.pezzelle@uva.nl

## Abstract

Animacy—whether an entity is alive and sentient—is fundamental to cognitive processing, impacting areas such as memory, vision, and language. However, animacy is not always expressed directly in language: in English it often manifests indirectly, in the form of selectional constraints on verbs and adjectives. This poses a potential issue for transformer language models (LMs): they often train only on text, and thus lack access to extralinguistic information from which humans learn about animacy. We ask: how does this impact LMs' animacy processing—do they still behave as humans do? We answer this question using open-source LMs. Like previous studies, we find that LMs behave much like humans when presented with entities whose animacy is typical. However, we also show that even when presented with stories about atypically animate entities, such as *a peanut in love*, LMs adapt: they treat these entities as animate, though they do not adapt as well as humans. Even when the context indicating atypical animacy is very short, LMs pick up on subtle clues and change their behavior. We conclude that despite the limited signal through which LMs can learn about animacy, they are indeed sensitive to the relevant lexical semantic nuances available in English.

## 1 Introduction

Animacy plays a significant role in cognitive processing, as evidenced by the fact that animate entities are easier to remember and prioritized in visual processing (Nairne et al., 2013; New et al., 2007; Bugaiska et al., 2019). It is so important that even young children can distinguish animate and inanimate entities (Rakison and Poulin-Dubois, 2001), and these are processed in distinct domain-specific brain regions (Caramazza and Shelton, 1998).

Animacy distinctions also manifest in language; however these distinctions may appear indirectly. While some languages explicitly mark animacy, animacy distinctions in English often take the form of selectional constraints that limit the use of certain verbs or adjectives with in/animate entities. For example, only animate entities can *walk* or *think*. So, while animacy is a rich distinction at the cognitive level, at the linguistic level, its signal can be muted.

Today's pre-trained transformer language models (LMs), however, are trained only on linguistic input. If they are to learn to process animacy, they must thus do so only from its downstream effects in text, unlike humans, who use visual and physical stimuli. We therefore ask: do such LMs respond to animacy in language as humans do?

We answer this by treating LMs as psycholinguistic test subjects, probing how they react to violations of animacy-related selectional constraints. Like prior work (Warstadt et al., 2020; Kauf et al., 2022), we first study LMs' responses in scenarios involving typical animacy. In such situations, animacy is a simple mapping between an object (e.g. *a peanut*) and its usual animacy (inanimate). We find that like humans, LMs generally prefer sentences that respect animacy-related selectional constraints, assigning higher probabilities to such sentences.

Unlike prior work, we also study *atypical* animacy (Coll Ardanuy et al., 2020), where a typically inanimate object becomes animate. We draw on Nieuwland and van Berkum (2006), which measured human N400 responses in scenarios with atypically animate entities like *a peanut in love*. We compare LM surprisal to human N400 brain responses and find that like humans, LMs are initially surprised to encounter entities like *a peanut in love*, but quickly adapt, becoming less surprised. Stronger LMs are more able to replicate the large magnitude of human N400 reduction.

Given LMs' success at adapting to atypical animacy with a long context, we test them on short sentences about atypically animate entities, and measure the extent to which their outputs reflect this atypical animacy. We find that even with lim-

ited context, LMs adapt their output distribution, treating the entity as animate. We conclude that, despite training without the modalities that humans use to learn about animacy, LMs respond to shifting animacy in a surprisingly human-like way. Code for our experiments is available at https://github.com/hannamw/lms-in-love.

## 2 Related Work

### 2.1 Animacy in Language

Animacy in cognition is often framed as a gradient phenomenon (de Swart and de Hoop, 2018). In language, this often simplifies to a tripartite hierarchy (humans > animals > objects) or a binary (humans & animals > objects); entities are distinguished synactically or morphologically by their position therein (Comrie, 1989).

Animacy exists at both the type level (e.g. dogs > rocks) and the token level (e.g. a specific rock in a story might be animate, though rocks are typically not). Moreover, linguistic animacy is based not only on biology, but also on the speaker's closeness and empathy with the entity in question (Kuno and Kaburaki, 1977); thus a speaker might treat their dog as more animate than an unknown dog.

The precise effects of animacy in language vary cross-linguistically, from explicit animacy marking to more indirect effects as in English. The latter include not only strict animacy-based selectional constraints (Caplan et al., 1994), but also more subtle grammatical influences (Rosenbach, 2008; Bresnan and Hay, 2008). For example, animate entities are more often mentioned first in a sentence, even if doing so produces less common constructions, such as the passive (Ferreira, 1994).

Here, we focus on the human / inanimate object dichotomy, and the animacy-based selectional constraints thereby imposed; this strong contrast should produce easier-to-measure effects in LMs.

### 2.2 LMs as Test Subjects

We study the behavior of LMs by treating them as psycholinguistic test subjects, a popular approach. One such line of work analyzes LMs by using the probability they assign to a sentence as a proxy for acceptability judgments. Generally, such studies provide pairs of sentences, one acceptable and one not; LMs must assign the more plausible sentence a higher probability. This method has been used to study LMs' processing of negation, subject-verb agreement, and more (Ettinger, 2020; Linzen et al.,

2016; Warstadt et al., 2020; Sinclair et al., 2022).

Other work compares LMs to humans by using surprisal—the negative log probability of a string—to estimate measures of cognitive effort during text processing. LM surprisal is versatile, and well-correlated with reading times, eye-tracking fixations, and EEG responses (Smith and Levy, 2013; Aurnhammer and Frank, 2018; Michaelov and Bergen, 2020); moreover, surprisal from stronger LMs provides better predictive power (Goodkind and Bicknell, 2018; Wilcox et al., 2020).

We use LM surprisal to predict the N400 brain response, which is elevated at semantically unusual content, like animacy-related selectional constraint violations. Studies have found that a word's surprisal correlates with human N400 response thereto (Frank et al., 2013, 2015; Michaelov et al., 2022); transformer LMs are the state of the art for this (Merkx and Frank, 2021; Michaelov et al., 2021).

### 2.3 Animacy Detection

We note that our interest in the processing of atypical animacy parallels similar developments in the NLP task of *animacy detection*: determining whether a given entity is animate. While many animacy detection studies originally considered only typical animacy (Orasan and Evans, 2007; Bowman and Chopra, 2012), later animacy detection work has recognized that entities' animacy may not always be typical, and may change over the course of a narrative (Karsdorp et al., 2015; Jahan et al., 2018). Particularly relevant for the present study, Coll Ardanuy et al. (2020) combine LMs and atypical animacy by using BERT for atypical animacy detection. Although this approach uses LMs as a tool to label animacy, rather than studying how LMs process animacy, we share their interest in the atypical edge cases of animacy.

### 2.4 Animacy in LMs

How neural models capture animacy has long interested cognitive scientists; Elman (1990) trained a simple neural LM on artificial language data, and found that its representations of animate and inanimate entities formed distinct clusters. More recent work has assessed the animacy-processing capabilities of modern LMs, mostly focusing on typical animacy. Animacy is one area tested by BLiMP (Warstadt et al., 2020), which we revisit in Section 4. Kauf et al. (2022) investigate animacy as part of LMs' generalized event knowledge; they also find that LMs are sensitive to (typical) ani-

macy as it pertains to selectional constraints.

We move beyond typical animacy to atypical animacy by using LM surprisal to replicate Nieuwland and van Berkum's (2006) studies on human N400 response to atypical animacy. Contemporaneous work (Michaelov et al., 2023) replicates one of these experiments in the original Dutch. In contrast, we replicate all experiments from Nieuwland and van Berkum (and Boudewyn et al. (2019)). These highlight situations in which models can capture general trends, but fail to capture low-level nuances. Moreover, by studying a diverse set of English LMs, we can identify how LMs' strength affects their predictive power.

## 3 Models

We experiment with these models: GPT-2 small, medium, large, and XL (Radford et al., 2019); OPT 2.7B, 6.7B, and 13B (Zhang et al., 2022); and LLaMA 7B, 13B and 30B (Touvron et al., 2023).[1] We use autoregressive LMs, as we need to compute probabilities for whole sentences. Moreover, we choose open-source models, to make our work replicable. We provide implementation details in Appendix A.

## 4 Typical Animacy

We test models' responses to animacy in situations where the animacy of a given token, or instance of entity, aligns with the animacy of its type more generally (e.g. cats are animate; rocks are not).

**Experiment** We test the models in Section 3 on the *animate-transitive* and *animate-passive* datasets of the BLiMP benchmark (Warstadt et al., 2020). Each dataset contains 1,000 minimal pairs of synthetic English sentences that differ only by one or two words (Table 1). By construction, one sentence respects animacy constraints; the other violates them. We evaluate models on these datasets by computing the probability it assigns to each sentence of each minimal pair. A model gets an example correct if it assigns higher probability to the sentence that respects the animacy constraint. We compute model accuracy over each dataset.

**Results** Figure 1 displays results for each model. It also includes human baselines, reported directly from Warstadt et al. (2020), which indicate the proportion of examples where annotators preferred

---

[1]The names of OPT and LLaMA models indicate (approximate) parameter counts; the GPT-2 models have 117M, 345M, 762M, and 1.5B parameters respectively.

|   | Acc? | Sentence |
|---|------|----------|
| **T** | Yes | **Naomi** had cleaned a fork. |
| **T** | No | **That book** had cleaned a fork. |
| **P** | Yes | Lisa was kissed by the **boys**. |
| **P** | No | Lisa was kissed by the **blouses**. |

Table 1: BLiMP examples: we provide one example each from the **T**ransitive and **P**assive datasets. Each is a minimal pair of sentences: one **Acc**eptable and one not.

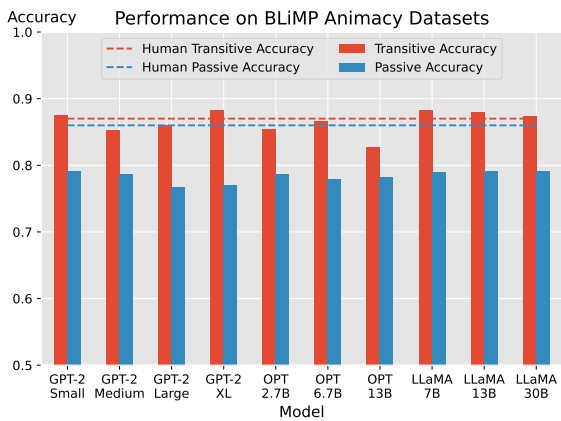

Figure 1: Model accuracy on BLiMP. Models match humans in the transitive, but not passive setting.

the acceptable sentence of the given minimal pair. Random performance for both datasets is 50%.

Models attain high performance in both scenarios. On transitive examples, they reach over 80% accuracy; some models prefer the sentence that respects animacy constraints more often than humans do (>87%). In the passive scenario, the gap between models (80%) and humans (86%) is wider.

This difference between the transitive and passive cases may be due more to setup differences than distinct animacy processing in the two scenarios. In the passive case, the target word is always in the last position, so model performance is determined only by the target's probability. In contrast, the target word is not the final token in the transitive case, so model success is determined by the probability of a longer string.

**Discussion** Our results indicate that models respect animacy constraints in typical scenarios: they match human performance on the transitive dataset, and are close behind on the passive. However, this test cannot distinguish between a model that truly understands animacy, and one that just associates words (types) with other words that reflect that word's typical animacy. For example, the model might simply associate a word like "shoe" with

verbs that take inanimate objects, without understanding that the inanimacy of an individual shoe is what prohibits its use with animate-selecting verbs.

To solve this problem, our analysis must move beyond type-level animacy, and test models' processing of animacy at the token level. We thus test models' responses to entities whose token-level animacy is atypical, distinct from their usual type-level animacy. If models process these entities according to their type-level animacy, their understanding of animacy is rather shallow. In contrast, models that process entities according to their token-level animacy may better understand animacy in full.

# 5 Atypical Animacy

In this section, we attempt to determine if LMs can capture animacy not only at the type-level, but also at the token-level. We do so by comparing model and human responses in cases of atypical animacy, where entities' canonical type-level animacy and their actual token-level animacy differ.

For human data, we turn to two similar studies—Nieuwland and van Berkum (2006) and Boudewyn et al. (2019)—that relied on the N400, a brain response measured via EEG that is elevated when processing semantically anomalous input. Both studies measured participants' N400 responses while they read stories where a typically inanimate entity acted as animate (Figure 2), similar in tone and content to a cartoon, or fairy-tale. Both found that while participants were initially surprised by the atypically animate entity, they quickly adapted, yielding low N400 responses to the entity.

We ask if the same is true of pre-trained LMs: can they adapt to entities that are animate at the token-level, despite being typically inanimate? Or is their processing of animacy limited to a simple type-level understanding? To answer this question, we replicate these studies with pre-trained LMs, using their surprisal to model N400 responses.

We replicate three experiments: Nieuwland and van Berkum's **repetition experiment**; their **context experiment**; and Boudewyn et al.'s **adaptation experiment**.[2] For each, we first explain the original study. Then, we explain how we adapt the experiment for LMs. Finally, we report our results and compare them the original study's results.

A nurse was talking to the **sailor/oar [1]** who had been in a violent boating accident. The sailor/oar cried for a long time over the storm that had raged over the lake for hours. The nurse consoled the **sailor/oar [3]**, saying that he would soon be well again. The sailor/oar complained of a bad headache that would not go away. The nurse gave the **sailor/oar [5]** a large dose of aspirin. The sailor/oar thanked her and fell asleep.

Figure 2: Story from Nieuwland and van Berkum, repetition experiment (translated and edited). Times when N400 responses were recorded are numbered, in bold.

## 5.1 Repetition Experiment

**Original Study**  In Nieuwland and van Berkum's first experiment, participants listened to Dutch stories that contained either a typical, animate entity or an inanimate entity behaving as if it were animate (Figure 2). Participants' N400 responses were measured at the 1st, 3rd, and 5th mentions of the entity (in Figure 2, either *oar* or *sailor*, in bold).

Nieuwland and van Berkum found that participants had a moderate N400 response to the first mention of a typically animate entity, and a low response on subsequent mentions. In contrast, participants initially had a high N400 response to the atypically animate entity. However, by the 3rd and 5th mentions thereof, their N400 responses were so low as to be statistically indistinguishable from the responses to mentions of the animate entity in the same contexts. Thus, while humans were initially surprised by the atypically animate entity, they quickly adapted to the situation, and found it no more surprising than typically animate entities.

**Our Experiment**  We model N400 responses with LM surprisal, as discussed in Section 2.2. For each of the 60 examples, we measure the surprisal of the animate and inanimate entity given the context at each timestep. For example, to model the inanimate N400 response at T1 in the example from Figure 2 given a model $p_\theta$, we compute $-\log_2 p_\theta(oar|A\ nurse\ was\ talking\ to\ the)$. Then, we compute the mean surprisal of examples containing animate and inanimate entities separately.

Since the original stimuli are in Dutch, we translate them to English, to make them compatible with the English LMs.[3] We do so using DeepL;[4] transla-

---

[2]In Appendix C, we replicate Boudewyn et al.'s English version of Nieuwland and van Berkum's context experiment with LMs; our results are identical to those of Section 5.2.

[3]We perform these experiments in Dutch in Appendix B. Dutch results are comparable to English results.

[4]https://www.deepl.com/translator

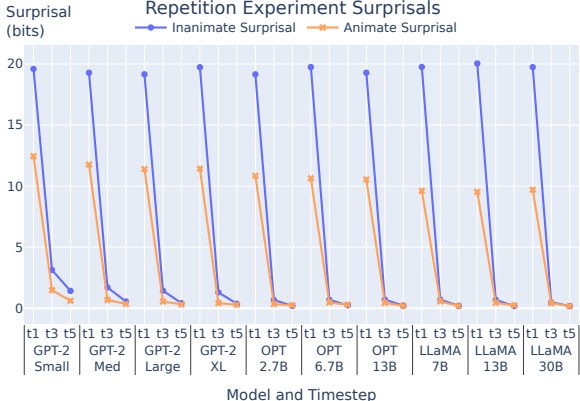

Figure 3: Mean repetition experiment surprisal. Inanimate surprisal is initially higher, but both surprisals decrease rapidly after T1, becoming near identical.

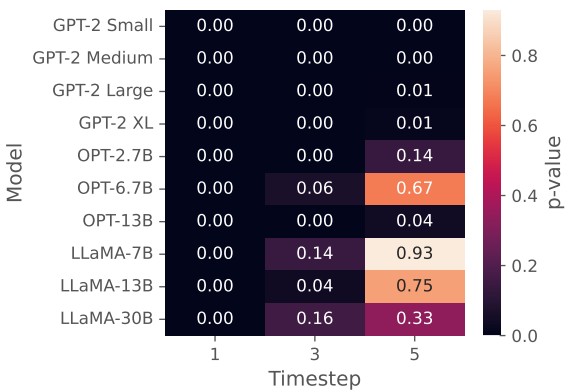

Figure 4: Stat. significance of the difference between animate and inanimate surprisal, by model and timestep

tions were checked by a native Dutch speaker. We then manually post-edited each stimulus to ensure it matched the cartoon-like tone and content of the original, and contained inanimate characters that violated typical animacy constraints in the 1st, 3rd, and 5th sentences of the stories.[5] Because we preserve the relevant aspects of the stimuli we expect the trends in N400 responses to be the same.

**Results** All models capture broad trends in human N400 responses well (Figure 3). At T1, models are very surprised by the inanimate entity, and only moderately surprised by the animate entity. At later timesteps, however, both entities' surprisals drop precipitously, to similar levels: models adapt to both entities quickly, just like humans do.

Still, the raw results do not prove that models

---

[5]We also edited stories for fluency, and to convert Dutch cultural references to Anglophone counterparts. The translated English stimuli can be found at `https://github.com/hannamw/lms-in-love`

A girl sat next to a diamond who was always doing strange things. The diamond told her that he liked to eat erasers. The girl ignored the diamond and his stories. Then the diamond said he also liked to sing songs. The diamond was quite **foolish/valuable** but secretly also very funny. That's why she always sat next to him.

Figure 5: Story from Nieuwland and van Berkum, context experiment (translated and edited). N400 responses were recorded at the words in bold.

are adapting to the extent that humans are. Since Nieuwland and van Berkum found that human N400 responses in the two conditions were statistically indistinguishable by T3, we test if the same is true for model surprisal. We use the Wilcoxon signed-rank test for non-normally distributed data (Wilcoxon, 1945) to determine if model surprisals for animate and inanimate entities are distinct at each timestep. We find (Figure 4) that like humans, LMs have a statistically significant difference between animate and inanimate surprisals at T1. However, while there was no difference in humans at T3, there are differences ($p < 0.01$) in most models; only the largest exhibit none. At T5, differences disappear in yet more large models. While models can generally approximate trends in human N400 responses to atypical animacy, only the largest and most powerful fully replicate human adaptation.

Overall, pre-trained LMs seem able to mimic human-like adaptation to atypically animate entities. It is tempting to conclude that they have a human-like understanding of animacy, that works at the token rather than the type level. However, it is equally possible that their decreased surprisal is due to repetition, rather than a deeper understanding of animacy. Transformer LMs even have a low-level emergent structure, induction heads, dedicated to such copy-pasting (Olsson et al., 2022).

Fortunately, Nieuwland and van Berkum shared this concern: humans might generate lower N400 responses only because they had seen the atypically animate token before. Thus, we also replicate their context experiment, which avoids this issue.

## 5.2 Context Experiment

Nieuwland and van Berkum's context experiment showed that participants' low N400 responses did not stem from lexical repetition.

**Original Study** As in the prior experiment, participants read 60 Dutch stories containing an atypically animate entity; at the end of each story, the entity was described using an adjective that was

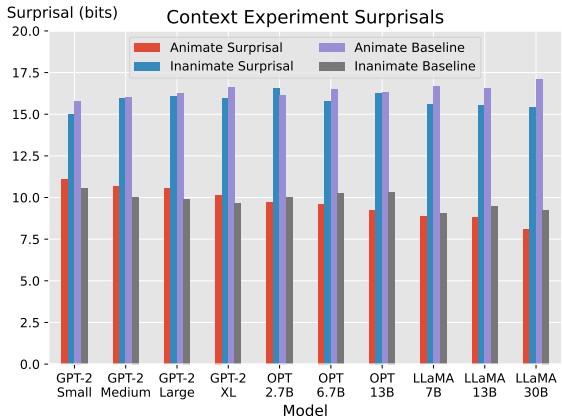

Figure 6: Context experiment surprisals. With context, the animate adjective is much less surprising; in the contextless baseline condition, this is reversed.

either context-appropriate (and generally used for animate entities) or context-inappropriate (but typical for the inanimate entity; Figure 5). The N400 response was measured at the adjective at the end of the story (*foolish* or *valuable*). N400 responses for the context-appropriate animate adjective were far lower than those to the entity-appropriate inanimate adjective, showing that the first experiment's effects were not caused by lexical repetition.

**Our Experiment** We calculate the surprisal at the animate and inanimate adjective for each of the 60 stories. We also compute baseline surprisals, the surprisal of the inanimate adjective without the entire story context, to show they are indeed high: e.g. $-\log_2 p_\theta(foolish|The\ diamond\ was\ quite)$.

**Results** For all models, mean surprisal of the animate adjective is much lower than that of the inanimate adjective (Figure 6). This is significant in all cases ($p < 0.01$; Wilcoxon signed-rank test). Moreover, the animate adjective is assigned higher probability in almost all cases—over 90% for large models. This mirrors the human trend: the N400 response to the context-appropriate animate adjective was much lower than the response to the entity-appropriate inanimate adjective. In the contextless baseline situation, the inanimate adjective receives a much lower surprisal than the animate adjective. Like humans, models use context and overcome their lexical knowledge regarding the traits can apply to animate and inanimate entities.

## 5.3 Adaptation Experiment

We now replicate Boudewyn et al.'s adaptation experiment, which combines the strengths of both

A lucky fellow/peanut had a big **smile [1]** on his face. The fellow/peanut was **amazed [2]** about his good fortune. Just now he had won the jackpot of two million dollars. The fellow/peanut was elated/salted and who could blame him.

Figure 7: Story from Boudewyn et al. (2019). N400 responses were recorded at the words in bold.

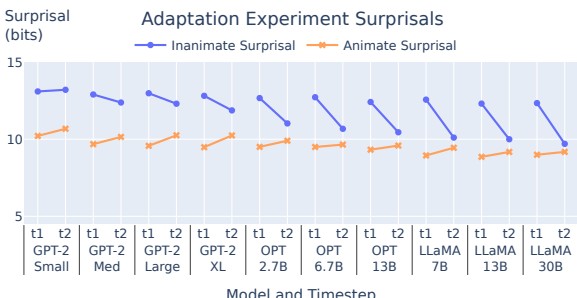

Figure 8: Adaptation experiment surprisals. Inanimate surprisal starts higher; the gap shrinks for larger LMs.

prior experiments. Like the first, it captures adaptation over time; like the second, it avoids the potential issues of repetition.

**Original Study** Boudewyn et al.'s adaptation experiment parallels the two previous experiments. Participants listened to 120 English-language stories containing either a typically or atypically animate entity (Figure 7). Participants' N400 responses were measured at the first content verb of the first two sentences of each story. These verbs signal that their subject is (perhaps atypically) animate, although they are notably not the same in each sentence. Findings mirrored those of Nieuwland and van Berkum: there was a sharp drop in N400 response at between the two timesteps in the inanimate scenario.

**Our Experiment** For each of the 120 stories, we calculate the surprisal at the two critical verbs, in the animate and inanimate case.

**Results** The results of the adaptation experiment (Figure 8) might appear starkly different from those of the repetition experiment. As before, surprisal at the critical word drops in the inanimate case, though only slightly. But in the animate case, surprisal remains constant or increases.

These results are in fact consistent with our earlier findings. The inanimate surprisal drop indicates that the short context sufficed to convince models of the entity's animacy. Moreover, the fact that surprisal does not drop in the animate case suggests that models are reacting specifically to the contex-

tual cues that the inanimate entity is animate, as opposed to the context more generally. The importance of model size is also consistent: stronger models have a smaller gap between animate and inanimate surprisals at T2. Although this gap is still significant for current models, trends indicate that stronger models may eliminate it.

## 5.4 Discussion

Across experiments, models replicate broad trends in human N400 responses. Stronger models replicate human results better, with lower surprisals at inanimate entities. Do they thus process animacy more like humans? We caution that low surprisals are not always desirable for cognitive modeling, as surprisal from LMs can underestimate processing difficulty in terms of reading time (van Schijndel and Linzen, 2021; Arehalli et al., 2022; Oh and Schuler, 2023); this may be because even relatively small LMs can predict next words as well as humans (Goldstein et al., 2022). Stronger LMs may only be better models of animacy processing in this situation because lower surprisals are desirable.

Regardless of the effects of model size, another question remains: do these positive results indicate that these LMs understand animacy? We cannot be certain: there exist mechanisms by which transformer LMs could perform well without any deep understanding of animacy. In the repetition experiment, models could use a copying mechanism to reduce their surprisal at repeated entities. In the context and even adaptation experiment, models could rely on the context, while ignoring the inanimate entity. This is a real concern: Michaelov et al. (2023) construct a (simple) model that does this.

In both cases, context is the complicating factor: LMs might exploit shallow context cues to simulate animacy processing effects, without having any real internal model of animacy. To investigate this question further, we study LMs' reactions to atypically animate entities in a low-context setting.

## 6 Low-Context Atypical Animacy

The previous experiments have shown that LMs can adapt in scenarios with atypically animate entities; however, LMs could have exploited shallow context features to do so, without any specific understanding or representation of animacy. We now investigate the extent to which LMs can leverage cues in the context by testing their behavior on very short sentences exhibiting atypical animacy.

**Dataset**  We craft a set of short incomplete sentences that describe an atypically animate entity (Table 2). The sentences are designed to elicit a critical next word—an adjective or a verb—that indicates if the LM treats the entity as animate. For example, if an LM continues "The boat snored and started to" with the verb "dream" this indicates that the boat is animate; continuing with "sink" does not. Unlike in prior experiments, these sentences provide only one clue indicating atypical animacy.

We create this dataset by defining prompts and filling them with nouns and verbs we sample from a predefined set. We sample from 181 nouns that humans rated as not very animate, but highly concrete; non-concrete inanimate nouns (e.g. "fear") cannot become animate, except metaphorically. We use concreteness ratings from Wilson (1988), and animacy ratings from VanArsdall and Blunt (2022).

For the verbs, we use a manually-filtered set of 191 verbs that imply their subject is animate, from Ji and Liang (2018). Each verb implies that its subject is animate for either psychological or physical reasons; e.g. *think* is psychological while *walk* is physical. Each verb co-occurs with human subjects at a high, high-mid, or mid frequency. We create a dataset of 10,000 items (Table 2) by sampling prompts, nouns, and verbs.[6]

**Experiment**  We run all LMs on the dataset. For each example, we evaluate whether the LM treats the entity in that example as animate by comparing the LM's output distribution to reference distributions. If our original sentence is $O =$"The chair spoke and began to", our inanimate reference is $I =$"The chair began to", while our animate reference is $A =$"The [human] began to", with a human entity randomly sampled from *person*, *man*, *woman*, *boy*, *girl*, and *child*. We indicate via $D_{KL}(A||O)$ the divergence between the next-word distributions given $A$ as context, and given $O$ as context, with other KL divergences defined analogously. We focus in particular on $D_{KL}(A||O)$ as **animacy divergence**; lower animacy divergence implies a more "animate" distribution.

**Quantitative Analysis**  Figure 9 shows KL divergences between the atypically animate sentence ($O$) and the reference distributions ($A/I$). For all models, the divergence between the inanimate

---

[6]A list of prompts, nouns and verbs is in Appendix F. Full dataset is available at https://github.com/hannamw/lms-in-love. For further experiments that vary properties of this dataset, see Appendix D.

| Sentence | Rank | #1 | #2 | #3 | #4 | #5 |
|---|---|---|---|---|---|---|
| The kilt commented and started to | 1 | walk | get | move | laugh | say |
| The cart noticed and was very | 3 | angry | ups | excited | interested | nerv |
| The dime retired and was very | 9994 | rare | valuable | scar | popular | collect |
| The telephone sighed and began to | 10000 | ring | v | bu | speak | be |

Table 2: Dataset examples and their top-5 continuations (sometimes partial words). The example at rank $n$ has the $n$th lowest animacy divergence (of 10,000). Low-divergence examples have animate continuations; high-divergence ones are stereotypical and inanimate. The top two examples use psychological verbs; the bottom two, physical.

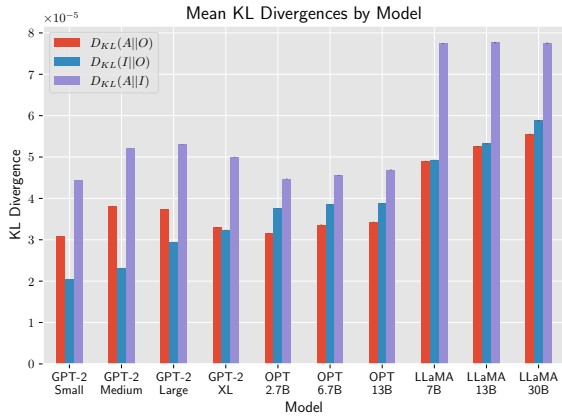

Figure 9: KL Divergence between atypically animate ($O$) and animate ($A$) / inanimate ($I$) references. Error bars (95% CI) are marked, but extremely small. The lower the bar, the more similar the distributions.

and animate references (purple) is the highest. Although only the added verb separates the atypically animate sentence from the inanimate reference, this leads its divergence with the animate reference (red) to be consistently lower; that is, adding the verb significantly increased the distribution's animacy. For OPT models, the animacy divergence is the lowest of their three divergences: the atypically animate distribution is even more similar to the animate distribution (red) than inanimate distribution (blue). This trend holds true for LLaMA models as well, though only weakly. Still, all model behavior clearly shifts with the addition of the animacy-implying verb.

To understand the cause of this shift, we analyze the data with respect to the known factors that vary between our prompts, to discern which affected model behavior. Results were similar across models, so we display results for one model, LLaMA-7B.

We first analyze the effect of our prompts, focusing on the difference between those that elicit verbs ("and began to...") and those that elicit adjectives ("and was very..."). We find (Figure 10, A) that verb-eliciting prompts produce lower animacy divergences than those that elicit adjectives.

We then analyze the effects of verbs and nouns on the sentence. For each verb or noun, we calculate the mean animacy divergence of sentences containing it. We observe a wide spread per verb and per noun (Figure 10, D and E), suggesting that they both impact model behavior. We then study the factors affecting individual nouns' and verbs' divergences. We find that psychologically animate verbs have somewhat lower divergence than physically animate verbs; psychologically animate verbs produce more animate behavior from LMs ($p < 0.01$, T-test; Figure 10 B). Sentences with verbs that had higher co-occurrence with humans had a lower divergence than those with a lower co-occurrence ($p < 0.01$; Figure 10 C),[7] though the effect size is rather small. For the nouns, however, neither animacy nor concreteness explains trends in animacy divergence.

**Qualitative Analysis** We also qualitatively verify that sentences with low animacy divergence have more animate continuations than those with high divergence. We sort examples by divergence and examine their top-5 continuations, focusing on examples at the top and bottom of the list.

Table 2 shows that animacy divergence aligns well with the qualitative animacy of continuations. Low-divergence examples have entities that *laugh*, or are *angry*. High-divergence ones have continuations stereotypical for the inanimate entity: a collectable *dime* becomes *rare*, or a *telephone* begins to *ring*.

**Discussion** Results show that LMs can adapt to atypical animacy even given limited cues: one verb, rather than an entire story. They also suggest that LMs do not require a long context to adapt; however, the factors that regulate this adaptation are complex. Some, like the choice of prompt, bear no clear relation to animacy; others, like the nouns,

---

[7]Significant for all three groups (F-test), and pairwise.

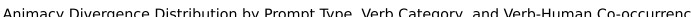

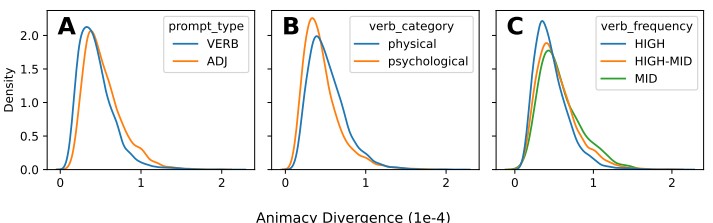
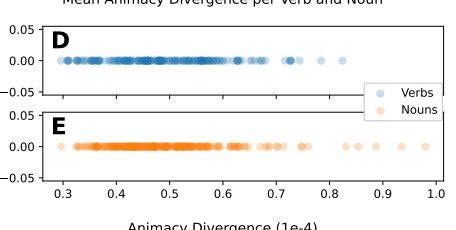

Figure 10: Left: Distribution of animacy divergences by prompt type, verb category, and verb-human co-occurrence. Right: Distribution of mean animacy divergences per-verb and per-noun. Each point is one verb (or noun).

show no clear pattern in how they affect model responses. Still, some interpretable factors exist. Psychological verbs may induce more animate continuations because they indicate animacy more strongly. While an inanimate object might metaphorically engage in physical activities like *dance*, they seldom *marry* or *volunteer*. These psychological words may thus serve as stronger signals of animacy.

## 7 Conclusions

Although animacy manifests only indirectly in English—it is not morphologically marked—pretrained LMs demonstrate relatively good animacy processing abilities. They both respect typical animacy and adapt to atypical animacy at close-to-human levels, although differences remain. They also demonstrate some ability to adapt to atypical animacy even when indicated by a very short context. LM adaptation still lags behind that of humans, but large models increasingly shrink the gap. We conclude that in the scenarios we test, LMs respond to animacy like humans do; however, our behavioral methodology can yield no conclusions about how models achieve this. Related work on world models (Li et al., 2021, 2023) suggests that by using causal techniques to search for internal structure in models, future work could not only demonstrate that LMs respond well to animacy, but also explain how they do so.

## Limitations

In this study, we use primarily behavioral experiments. These are suitable for comparing models to human data, but do not reveal the causal mechanisms by which LMs process animacy. In order to discover these, it would be more appropriate to use causal interventions or similar techniques, which we do not explore.

Considering the limitations of behavioral techniques, this study is still limited by the fact that it did not collect human data. We translate Nieuw-

land and van Berkum's stimuli to English, but do not test them again on native English speakers; the N400 responses to Dutch data could differ from N400 responses to English data, even though we assume they will be similar. Similarly, studying human responses to our low-context atypical animacy stimuli (Section 6) would better inform our analysis of LM performance.

## Ethics Statement

This work presents only minor ethical concerns. A particular concern is one of bias and stereotypes; the original stories in Nieuwland and van Berkum (2006) do contain stereotypes. We attempt to soften these in our translations, but some stereotypes are still present in the translated material.

More generally, LMs such as those analyzed must be used with caution. Although such models achieve high performance on language-based tasks, this performance does not necessarily stem from genuine linguistic understanding. Moreover, models can not only perpetuate harmful biases present in their training data, but also create misleading or false output.

## Acknowledgements

The authors thank the members of the Dialogue Modelling Group, particularly Joris Baan and Raquel Fernández, as well as Marianne de Heer Kloots, for their helpful feedback. They also thank the authors of the original human studies, in particular Mante Nieuwland and Megan Boudewyn, who allowed us to translate / use their stimuli. Finally, the authors thank participants of the ELLIS, AI4media, and AIDA Symposium on Large Language and Foundation Models in Amsterdam for their insightful comments. This work was supported by the ISRAEL SCIENCE FOUNDATION (grant No. 448/20), Open Philanthropy, and an Azrieli Foundation Early Career Faculty Fellowship.

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

# A    Implementation Details

We implement all experiments in PyTorch (Paszke et al., 2019). For all models but LLaMA, we use the implementations and weights publicly available via the HuggingFace Transformers library (Wolf et al., 2020); for LLaMA, weights are only available upon request via a form at `https://github.com/facebookresearch/llama`. We run models using an Nvidia A100 40GB GPU (or multiple when necessary, as for LLaMA 30B). The runtime of these experiments should not exceed 24 hours, even when run serially.

# B    N400 Results for Dutch Data

The methods for this set of experiments are mostly identical to those of the English experiments. However, instead of English LMs, we use Dutch LMs. We consider GPT-2 Small trained from scratch on Dutch, and GPT-2 Medium trained in English, with fine-tuned Dutch word embeddings (de Vries and Nissim, 2021), and GPT-2 Medium and Large trained from scratch on Dutch (Havinga, 2021). These represent the best Dutch autoregressive LMs available.

## B.1    Results: Repetition Experiment

The results of the repetition experiment in Dutch (Figure 11) are rather similar to those in English. As before, the surprisal starts out high for both

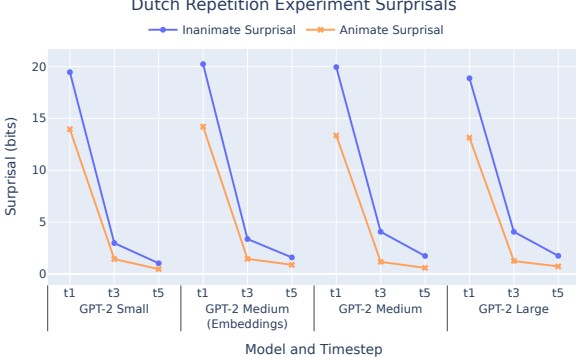

Figure 11: Repetition experiment surprisals. Surprisal drops rapidly after T1, with inanimate surprisal drawing close to animate surprisal.

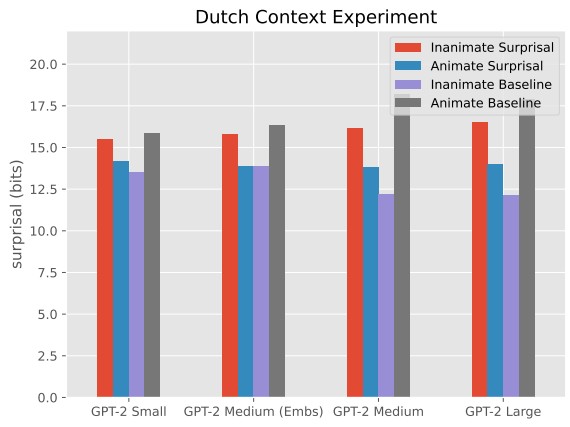

Figure 12: Context experiment surprisals. With context, the animate adjective is much less surprising; in the contextless baseline condition, this is reversed.

animate and inanimate entities (though higher for the latter). The difference between these two is less pronounced than in English. The surprisal drops rapidly at T3, and again at T5. Unlike in English, there are no strong model-wise trends, whereby stronger models have a lower difference in surprisals. And in all cases, the difference between the two conditions at T5 is statistically significant.

## B.2 Results: Context Experiment

Again, the results of the Dutch experiment (Figure 12) are much like the English results. Surprisals at animate adjectives are much lower than those at inanimate adjectives; however, in the baseline condition, which lacks context, the trend is reversed.

## C Boudewyn et al.: English Context Experiment

**Original Study** Much like in Nieuwland and van Berkum (2006), Boudewyn et al. (2019) also mea-

> A lucky peanut had a big smile on his face. The peanut was amazed about his good fortune. Just now he had won the jackpot of two million dollars. The peanut was **elated/salted** and who could blame him.

Figure 13: Story from Boudewyn et al. (2019), context experiment. N400 responses were recorded at the words in bold.

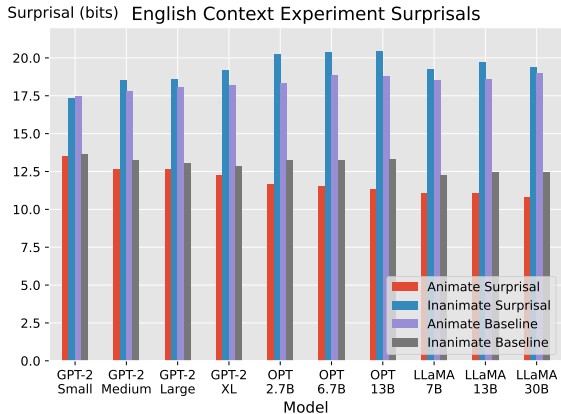

Figure 14: Surprisals for animate and inanimate adjectives in the normal and baseline condition

sured participants' N400 responses to adjectives at the end of each story (Figure 13). Each adjective was either typical for animate entities (and thus context-appropriate) or typical for the inanimate entity in question (but context-inappropriate). N400 responses were much lower in the former case than in the latter.

**Experiment** As in the earlier context experiment (Section 5.2), for each of the 120 stories, we calculate the surprisal at the animate and inanimate adjective. We also compute baseline surprisals, defined as the surprisal of the critical adjective given only the sentence containing it as context.

**Results** The results of the English context experiment (Figure 14) are like those of the earlier context experiment (Section 5.2). Like before, surprisal at the animate adjective is much lower than surprisal at the inanimate adjective; in the baseline condition, this trend is reversed.

## D Further Low-Context Experiments

In the following experiments, we made changes to our and experimental setup, in order to ensure that our findings were not a result of any idiosyncrasies of our dataset's construction.

## D.1 Larger Human Entity Sample Pool Experiment

In our original experiment, our human reference was $A =$"The [human] began to", where [human] was sampled from a very limited pool: *person*, *man*, *woman*, *boy*, *girl*, and *child*. The pool was limited for two reasons. First, we did not want to introduce another axis of variation across examples.

Second, and more importantly, we wanted to create a generic "human-like" action distribution. Some nouns (e.g. a musician, or a thief) are animate humans, but their distributions over next actions are probably skewed in ways that are not representative of animate entities in general; musicians are likely to play music, and thieves, to steal. Thus, their next-token distributions might have a high KL-divergence with those of atypically animate objects for reasons unrelated to the object's perceived animacy. However, we still wanted to ensure that our findings are not reliant on this small pool of humans. To verify this, we constructed a larger pool of human entities to sample from.

To construct this pool, we first added in generic human entities (*man*, *woman*, *person*, *boy*, *girl*, *child*, *teenager*) and family relations (*mother*, *father*, *grandfather*, *grandmother*, *wife*, *husband*, *grandchild*, *granddaughter*, *grandson*, *aunt*, *uncle*, *niece*, *nephew*, *cousin*). Then, we added more entities, starting from a list of job titles,[8] which are all naturally human nouns. We then used word frequency data from 1 million words of Wikipedia ([Goldhahn et al., 2012](#)), filtering out any job titles that are not found in that text. We then manually added valid job titles to our pool in order of descending frequency, until our expanded pool numbered 100 entries: 21 generic human entities and 79 professions. We could have increased the pool size by including more jobs, but the job title pool was noisy, and needed manual filtering to weed out nonsensical jobs, and jobs that sound too much like inanimate objects.

Then, we conducted our experiment again as in Section 6, using this larger pool. The results of this experiment (Figure 15) are rather similar to that of our original experiment. In all cases the output distribution $p_O$ is clearly more animate than $p_I$; in the OPT models, the $p_O$ is more like the animate ($p_A$, red) than inanimate ($p_I$, blue)

[8]The job titles originate from https://github.com/jneidel/job-titles/tree/master, which collects job titles from a variety of sources.

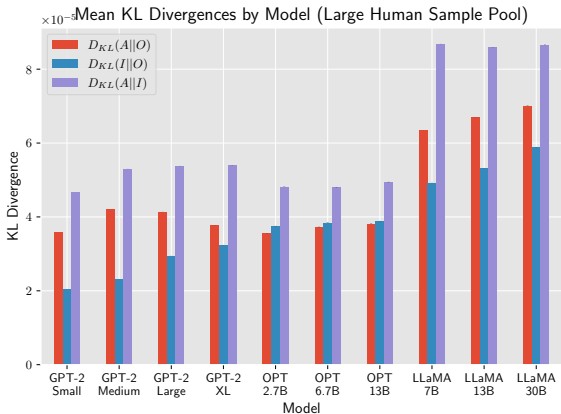

Figure 15: KL Divergence between atypically animate ($O$) and animate ($A$) / inanimate ($I$) references, where the animate entity in $A$ is drawn from a larger pool. The lower the bar, the more similar the distributions.

distribution. However, the animacy divergence is somewhat higher for the LLaMA models.

## D.2 Matched Frequency Human Entity Experiment

In this experiment, we ensure that differences between the frequency of the sampled human entity and that of the inanimate entity do not undermine our experimental setup. Using the same frequency data as before, we matched each inanimate entity in our pool to the (manually verified, valid) human entity with the most similar frequency. We excluded one inanimate object ("well") because its frequency was confounded by the very common adverb that shares its form. We were able to find a good human match for each inanimate object: the object frequency-to-human frequency ratio ranged from 0.92 to 1.09; this is near 1, the ideal ratio.

Then, we conducted our experiment again as in Section 6, using this larger pool. The results of this experiment (Figure 16) are strikingly similar to those of the previous experiment (Figure 15). We take this to suggest that frequencies are indeed not very important to the phenomenon we observe.

## D.3 Cataphoric Prompt Experiment

In our original experiment, we test sentences of the form $O =$"The chair spoke and began to". One potential concern is that LMs might just look at "spoke and began to", which implies an animate continuation, thus overcoming the inanimacy of *chair*. We can test this by using a cataphoric prompt, where a referring pronoun comes before the object, e.g. $O' =$"After it spoke, the chair began to". In such a prompt, there now exists the 4-gram

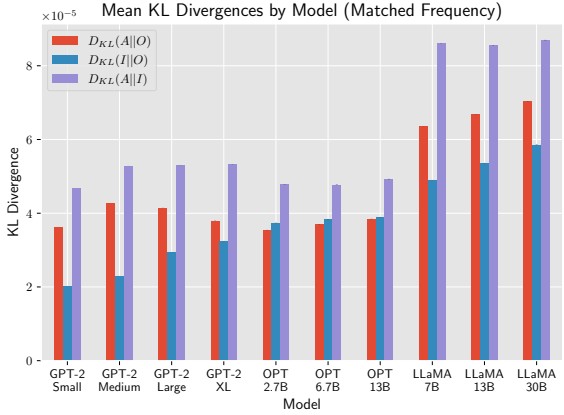

Figure 16: KL Divergence between atypically animate ($O$) and animate ($A$) / inanimate ($I$) references, where the animate entity in $A$ is matched in frequency with the inanimate entity in $I$ and $O$. The lower the bar, the more similar the distributions.

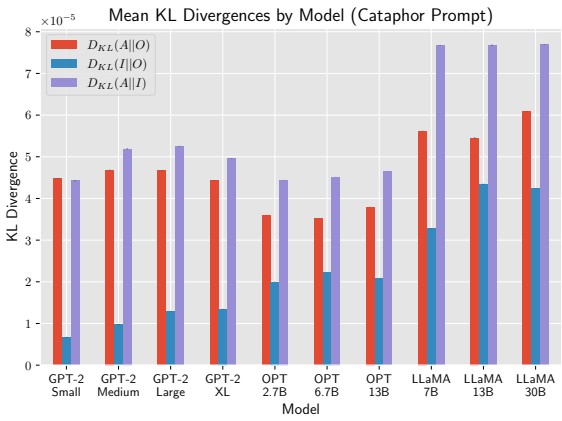

Figure 17: KL Divergence between atypically animate ($O$) and animate ($A$) / inanimate ($I$) references, using the cataphoric prompt. The lower the bar, the more similar the distributions.

"the chair began to", which should make this task a little harder for LMs. Using the less-animate / inanimate pronoun "it" to refer to the atypically animate object also makes this more challenging.

Then, we conducted our experiment again as in Section 6, using the original small pool of human entities. The results of this experiment (Figure 17) suggest that this setting is indeed harder for LMs; no longer to any LMs adapt such that divergence between $p_O$ and $p_A$ (red) is less than that between $p_O$ and $p_I$ (blue). However, there is still an increase in animacy compared to the case where the prompt does not hint at atypical animacy (purple).

# E  Low-context Animacy: Evaluation

We first tried to classify potential next tokens as animate, inanimate, or neither; we could then compute the probability assigned to each group. However, classifying all next tokens was noisy: even with resources like FrameNet or VerbNet (Baker et al., 1998; Kipper et al., 2000), it was infeasible to determine if a verb implied that its subject was animate.

# F  Low-Context Animacy Dataset Details

This section contains lists of the prompts, nouns, and verbs used in constructing this dataset. The full dataset, and all variants thereof, can be found at https://github.com/hannamw/lms-in-love.

## F.1  Prompts

- The [noun] [verb] and began to
- The [noun] [verb] and started to
- The [noun] [verb] and was very
- The [noun] [verb] and became very

## F.2  Nouns

accordion, ambulance, amplifier, appliance, arrow, automobile, axe, bagpipe, balloon, bandage, banner, barrel, basket, bin, biscuit, blanket, blossom, blouse, boat, bomb, book, bottle, bouquet, bra, bracelet, bread, brush, bubble, bucket, buckle, bullet, button, cake, camera, candle, candy, cane, cannon, canoe, cape, cart, casket, chisel, chocolate, clarinet, clock, clothing, coat, cocktail, coffin, coin, collar, corpse, dagger, dart, desk, dime, dress, engine, envelope, ferry, fiddle, firewood, flask, flute, football, fruit, furniture, glass, glove, goblet, gown, hailstone, hairpin, hammer, harp, hat, helmet, hose, jar, keg, kilt, knife, lamp, lantern, lens, limousine, mallet, map, mattress, medallion, microscope, mirror, missile, moccasin, nail, napkin, necklace, needle, nickel, nightgown, oar, ornament, oven, overcoat, pants, pearl, pencil, pendulum, penny, phone, photograph, piano, pie, pillow, pipe, plank, pot, propeller, prune, purse, quilt, radio, record, refrigerator, ribbon, rifle, ring, rope, rug, sandal, satchel, saxophone, scissors, scroll, shawl, shield, shirt, shoe, ski, skull, sleigh, sock, sofa, spoon, statue, steak, stove, submarine, sword, tablespoon, telephone, telescope, thermometer, thorn, thread, ticket, tie, timepiece, tractor, tray, tripod, trombone, truck, trumpet, tube, tweezers, twig, typewriter, umbrella, van, vase, vehicle, vest, violin, wallet, wheel, whistle, wig, yacht, zipper,

### F.3 Verbs

#### F.3.1 Physical

**High Co-Occurrence With Human Subjects**
stammer, grimace, mumble, drawl, frown, gasp,
yell, nod, smile, laugh, shrug, sob, grin, kneel,
wince, whisper, sigh, giggle, squint, murmur, doze,
fiddle, gesture, mutter, faint, gulp, flinch, chuckle,
drink, weep, stare, grunt, listen, watch, fumble,
shiver, pace, lean, blush, shout, gaze, walk, sit,
sleep, dine, pant, glare, clap, stumble, snore, shave,
wave, omit, sniff, piss, cough, wail, grumble,
breathe, snort, spit, eat, duck, die, swallow, growl,
blink, inhale, bellow, starve, crouch, yawn, step,
squat

**High-Mid Co-Occurrence With Human Subjects** pounce, scream, flee, shudder, wander,
shriek, stagger, wink, sing, whistle, jog, limp, hiss,
trot, jump, bathe, dance, paint, ramble, shower,
drown, recover, pack, sweat, bow, flush, crawl

**Mid Co-Occurrence With Human Subjects**
bark, swim, bleed, howl

#### F.3.2 Psychological

**High Co-Occurrence With Human Subjects**
think, know, wonder, remember, guess, exclaim,
retort, marry, notice, understand, hurry, pray, medi-
tate, swear, forget, enquire, realise, confess, apolo-
gise, hesitate, suspect, reply, talk, sneer, cry, dream,
moan, ponder, revel, learn, scowl, retire, snarl,
groan, speak, complain, beg, wait, preach, grieve,
read, plead, volunteer, answer, curse, choose, panic,
chant, cheat, salute, emigrate, protest, visit, lament,
misunderstand

**High-Mid Co-Occurrence With Human Subjects** consent, graduate, disagree, steal, mourn,
study, argue, search, insist, practise, interrupt, obey,
comment, concede, fight, applaud, enlist, worry,
teach, train, agree, struggle, rush, evacuate, object,
pay, pursue, hasten

**Mid Co-Occurrence With Human Subjects**
vote, invest, register