# OpenReview forum: "When Language Models Fall in Love: Animacy Processing in Transformer Language Models"
_EMNLP/2023/Conference — EMNLP 2023 Main_

### Official Review · Reviewer_QnBL · 2023-08-05

**Soundness:** 4

**Excitement:**

5: Transformative: This paper is likely to change its subfield or computational linguistics broadly. It should be considered for a best paper award. This paper changes the current understanding of some phenomenon, shows a widely held practice to be erroneous in someway, enables a promising direction of research for a (broad or narrow) topic, or creates an exciting new technique.

**Paper Topic And Main Contributions:**

This paper explores how transformer models process animacy. While previous work has shown that LMs are sensitive to type-level animacy, this work looks at cases with atypical token-level animacy, e.g. an oar that is treated as animate. Previous EEG data shows that humans are able to quickly adapt to these cases of atypical animacy, and this work shows that LMs also adapt quickly in similar cases.

**Questions For The Authors:**

Line 304: What does it mean to make stimuli cartoon-like?

**Reasons To Accept:**

This paper addresses a fresh topic that is interesting from both the vantage point of cognitive modeling and BERTology. The experiments are presented very clearly, and do well at anticipating potential pitfalls, e.g. that LMs might show decreased surprisal merely due to lexical repetition. The overall finding that LMs respond in humanlike ways to atypical animacy seems like a novel, non-trivial result.

**Reasons To Reject:**

The significance testing in Section 5.1 (described starting on line 321) seems sort of backwards. The real goal is to see if there is evidence that animate and inanimate surprisal are the same -- this would show that LMs are doing something special -- but the Wilcoxon test sets this as the null hypothesis. A negative result from a significance test does not support the null hypothesis. AFAIK, the proper kind of test to do here would be an equivalence test, where the alternative hypothesis would be that the surprisals match and the null hypothesis would be that they differ. The same concern applies to Section 5.3.

**Reproducibility:**

5: Could easily reproduce the results.

**Reviewer Confidence:**

4: Quite sure. I tried to check the important points carefully. It's unlikely, though conceivable, that I missed something that should affect my ratings.

---

> ### Author Rebuttal · Authors · 2023-08-28
>
> Thanks for your careful review! We’re happy to hear you found our work novel, and our presentation clear. Below, we’ve answered your two main points.
>
> > The significance testing in Section 5.1 (described starting on line 321) seems sort of backwards. The real goal is to see if there is evidence that animate and inanimate surprisal are the same -- this would show that LMs are doing something special -- but the Wilcoxon test sets this as the null hypothesis. A negative result from a significance test does not support the null hypothesis. AFAIK, the proper kind of test to do here would be an equivalence test, where the alternative hypothesis would be that the surprisals match and the null hypothesis would be that they differ. The same concern applies to Section 5.3.
>
> This is an interesting point. In principle, you are correct: the Wilcoxon test can only (fail to) reject the null hypothesis of equivalence; an equivalence test is needed to show that the two conditions yield the same result. However, the original human studies do not use equivalence tests. Instead, they too fail to reject the null hypothesis, concluding there is no statistically significant difference between the two conditions.
>
> Our study builds on the originals by comparing LM results to those of humans. Lacking results of equivalence tests run on human data, we would have nothing to compare LM equivalence test results to. For this reason, we did not perform such tests.
>
> We could perform equivalence tests anyway, but we run into a practical problem. If we take the two one-sided tests (TOST) approach to equivalence testing, we ought to set bounds on the difference between our two conditions. Ideally, these bounds would reflect the bounds used by the corresponding tests on human data, but we have none. Alternatively, we should set our bounds such that any difference falling within the bounds is not meaningful, but what this means in terms of surprisal is not obvious. We could get around this by using something like Cohen’s d to determine the magnitude of surprisal difference that constitutes a meaningful effect size; however, this is complicated by the fact that Cohen’s d relies on a normality assumption, which our data violate.
>
> In the interest of exploring equivalence tests, we performed equivalence tests (TOST, Wilcoxon signed-rank), and found that equivalence holds for some reasonable bounds, chosen to exclude small effect sizes as measured by Cohen’s d. However, the results always depend on the precise bounds chosen. We thus do not want to make any claims about equivalence, or how this might speak to LMs’ processing of animacy with respect to humans’.
>
> This was a well explained and subtle point, so we thank you for raising it. If you have a clearer idea of how to choose bounds, or how to link equivalence test results to the prior human results, please let us know!
>
> > Line 304: What does it mean to make stimuli cartoon-like?
>
> Cartoon-like stimuli is a phrase used by Nieuwland and van Berkum (2006). It refers to the fact that the original stimuli were humorous stories about concrete inanimate objects that are alive, as might often happen in a cartoon; we aimed to maintain these qualities. We’ll revise the paper to clarify this.

---

### Official Review · Reviewer_w2ux · 2023-08-05

**Soundness:** 4

**Excitement:**

4: Strong: This paper deepens the understanding of some phenomenon or lowers the barriers to an existing research direction.

**Paper Topic And Main Contributions:**

This paper describes experiments showing large language models seem to be able to detect animacy in text, even with very little information.

**Reasons To Accept:**

The paper considers an interesting semantic question and uses what seems to me to be a reasonable experimental design to evaluate it.

**Reasons To Reject:**

I only have a few minor issues.

First, I'm not sure why the expression of animacy in text is called 'muted' (in the introduction) and 'indirect' (in the conclusion).  Animacy is a major feature in causatives, for example, so I was not surprised to see it is learnable from data.

Second, examples like 'The boat snored and started to _' in the third experiment might be confounded by the recency of the conjoined animate verb (in this case, 'snored').  I wonder if this could be mitigated by using cataphoric stimuli like 'After it snored, the boat started to _' that place the inanimate noun closer to the mask.

Finally, no significance testing was reported for results in Section 4.


**Reproducibility:**

4: Could mostly reproduce the results, but there may be some variation because of sample variance or minor variations in their interpretation of the protocol or method.

**Reviewer Confidence:**

4: Quite sure. I tried to check the important points carefully. It's unlikely, though conceivable, that I missed something that should affect my ratings.

**Typos Grammar Style And Presentation Improvements:**

* The word 'entity' is repeated in Section 5.3.
* In Section 6, I had trouble finding the antecedent of 'they' in the sentence beginning 'We find that psychologically animate verbs ...'.

---

> ### Author Rebuttal · Authors · 2023-08-28
>
> Thanks for your thoughtful review! We’ve answered your questions below.
>
> > First, I'm not sure why the expression of animacy in text is called 'muted' (in the introduction) and 'indirect' (in the conclusion). Animacy is a major feature in causatives, for example, so I was not surprised to see it is learnable from data.
>
> Our main motivation for describing animacy as “indirect” in English is that it isn’t explicitly morphologically marked, in contrast with other phenomena (e.g. number), or animacy in other languages (e.g. Slavic languages). We will clarify this in the introduction!
>
> > Second, examples like 'The boat snored and started to _' in the third experiment might be confounded by the recency of the conjoined animate verb (in this case, 'snored'). I wonder if this could be mitigated by using cataphoric stimuli like 'After it snored, the boat started to _' that place the inanimate noun closer to the mask.
>
> This is an interesting question! We performed preliminary tests on this new prompt and found that our results still hold: animacy divergence is still reduced, compared to divergence between the animate and inanimate distributions. The effect size is smaller, but overall trends, such as stronger language models having generally lower animacy divergence are the same. We found the same results for the prompt “After snoring, the boat started to”, which we tested to avoid the use of “it”, which could have suggested a lower level of animacy than “he” or “she”, and cued LMs towards an inanimate continuation. We’ll add these new results to the paper!
>
> > Finally, no significance testing was reported for results in Section 4.
>
> This is true: we did not report significance tests. If you are referring to significance tests between model and human performance, this is due to a lack in human data. BLiMP’s human annotations are sparse: for each dataset (passive and transitive), 5 examples are annotated, with 20 human judgments each. So, paired statistical tests are impossible, and even the human accuracy metric is based on a very small number of examples, making it less than reliable. For this reason, we forgo statistical tests, and only broadly check that model and human performances are similar, before moving on to the main contributions of our paper.

---

### Official Review · Reviewer_9t4f · 2023-08-10

**Typos Grammar Style And Presentation Improvements:** not only not only -> not only
**Soundness:** 4

**Excitement:**

4: Strong: This paper deepens the understanding of some phenomenon or lowers the barriers to an existing research direction.

**Missing References:**

Previous work on animacy detection, e.g.

- Ardanuy et al (2020) who focus on atypical animacy
- Jahan et al (2018)

**Paper Topic And Main Contributions:**

The paper presents a study of animacy processing in large language models, comparing it to human processing as evidenced by psycholinguistic studies. The paper analyzes animacy processing in openly available datasets, such as BLIMP, however, the main contribution is found in the adaptation of several Dutch psycholinguistic studies to evaluation by English LMs.

**Questions For The Authors:**

* While the focus in this paper is clearly English, given that the original study was performed in Dutch, it might be interesting to know a bit more about animacy processing in other languages?

* What about words that are ambiguous at the type level, e.g. metonyms like "The White House"? Would be interesting to compare their processing to the clear-cut cases used in this study.

* Who performed the psycholinguistic experiments under "Adaptation Experiment"? This is unclear from the text, as it can be read as being the authors and not Nieuwland and van Berkum. Please clarify.

* For the low-context scenario, it seems unfortunate that the human entity for the A reference is randomly sampled, rather that being matched in terms of frequency with the inanimate noun. Did you consider this? Could frequency be influencing the processing here?


**Reasons To Accept:**

- The paper is very well-written and clear, motivating each study well and guiding the reader through the results in an admirable manner.
- The paper further navigates its interdisciplinary setting very well, by explaining the psycholinguistic studies and describing in enough detail its adaptation to LM evaluation.
- The paper makes several contribution both in terms of datasets (which they will release) and experimental results.

**Reasons To Reject:**

There are only a few minor issues:
- the background section is missing previous work on animacy detection, which, while somewhat different from the current experimental setup should still be mentioned (see below)
- it is unclear exactly who performed the final psycholinguistic experiment, might just be a missing reference, but should be easy enough to clarify by the authors

**Reproducibility:**

4: Could mostly reproduce the results, but there may be some variation because of sample variance or minor variations in their interpretation of the protocol or method.

**Reviewer Confidence:**

4: Quite sure. I tried to check the important points carefully. It's unlikely, though conceivable, that I missed something that should affect my ratings.

---

> ### Author Rebuttal · Authors · 2023-08-28
>
> Thank you for your review! We’re glad that you found the interdisciplinary aspects of our work to be clearly presented. We answer your questions below, point by point.
>
> > the background section is missing previous work on animacy detection, which, while somewhat different from the current experimental setup should still be mentioned (see below)
>
> It’s true that our related work section omits animacy detection; this was cut in favor of discussing animacy and behavioral interpretability. If accepted, we’d be happy to use the extra page given to camera-ready papers to include the papers you’ve cited.
>
> > Who performed the psycholinguistic experiments under "Adaptation Experiment"? This is unclear from the text, as it can be read as being the authors and not Nieuwland and van Berkum. Please clarify.
>
> The original human adaptation experiment comes from Boudewyn et al. (2019), while the two prior experiments (repetition and context, in Sections 5.1 and 5.2) come from Nieuwland and van Berkum (2006). We will revise our paper to make this even more explicit!
>
> > While the focus in this paper is clearly English, given that the original study was performed in Dutch, it might be interesting to know a bit more about animacy processing in other languages?
>
> We look into Dutch models in Appendix B. Trends for Dutch models are quite similar to trends for English models: they show a large reduction in surprisal at the 3rd timestep (repetition experiment, 5.1/B.1), and prefer the animate over inanimate adjective (context experiment, 5.2/B.2). In general, higher-performing Dutch LMs showed larger reductions in surprisal, and thus more human-like performance. However, no Dutch LMs were able to replicate phenomena such as the lack of difference between animate / inanimate surprises in the repetition experiment. We attribute this to the lack of truly high-performing Dutch autoregressive LMs, given that better models yielded more human-like surprisals.
>
> With respect to human animacy processing, the similarity between the findings of Nieuwland and van Berkum (Dutch) and Boudewyn et al. (English) suggests that in this case, humans also process atypical animacy in a similar way across the two languages.
>
> > What about words that are ambiguous at the type level, e.g. metonyms like "The White House"? Would be interesting to compare their processing to the clear-cut cases used in this study.
>
> This is interesting! Intuitively, it seems like the degree to which such a word is treated as animate should be related to the frequency with which it is used in its metonymic sense. It would be valuable to investigate if models exhibit this sort of animacy gradient. This question deserves a thorough investigation which is outside the scope of our current work; however, we would love to see further exploration of this topic, and hope that future work can build on ours to answer it.
>
> > For the low-context scenario, it seems unfortunate that the human entity for the A reference is randomly sampled, rather that being matched in terms of frequency with the inanimate noun. Did you consider this? Could frequency be influencing the processing here?
>
> We generally don’t think that frequency is influencing processing in this scenario. If you were suggesting that, for each inanimate entity, the corresponding human entity be set to a human entity with the same frequency, this seems feasible. However, we don’t think the frequency of the human entity, relative to that of the inanimate noun, should affect our results. The human and inanimate entities never appear in the same sentence; we only compare their distributions. So, any frequency effects would be somewhat indirect.
>
> We run such experiments just to be sure, and find that the results in the matched and unmatched cases are the same. Our procedure was as follows: we constructed a list of humans from a list of professions, as well as a handcrafted list of common family relations. Then, we filtered out those humans that didn’t appear in our frequency corpus. We excluded one inanimate object (well) because its frequency was confounded by the very common adverb that shares its form. We were able to find a good human match for each object—the object freq.-to-human freq. ratio ranged from 0.92 to 1.09, near 1, which is the ideal ratio.
>
> We then ran the low-context experiments in both the matched case, where each object was compared with its paired noun, and the unmatched case, where each object was paired with a randomly sampled noun. Note that the pool from which samples were taken in the unmatched case, contains only those human entities that were matched with an object in the matched case. Thus, the overall pool of human entities is the same; only the matching differs. The results of the two experiments are the same, and align with our original experimental results.

---

### Official Review · Reviewer_jHyH · 2023-08-10

**Soundness:** 4

**Excitement:**

4: Strong: This paper deepens the understanding of some phenomenon or lowers the barriers to an existing research direction.

**Paper Topic And Main Contributions:**

When Language Models Fall in Love: Animacy Processing in Transformer Language Models

This paper uses a series of experiments in order to compare human and LMs reactions to animacy processing.
The authors used existing experiment designs for humans (with recorded results) to test LMs.
The authors used existing datasets and translated them from Dutch to English.
The authors also added their own experiments.

Finally, the authors conclude that according to the experiments, LMs generally respond to animacy like humans.

**Questions For The Authors:**


A: Can you slightly lower the tone of the conclusion?

**Reasons To Accept:**


I enjoyed reading the paper. The topic is interesting, and the writing is good.
Each experiment is described clearly and contains relevant and interesting discussions.

I liked the use of existing tests from the literature and the comparison with human results.

**Reasons To Reject:**


Despite the findings and results of the experiments, the conclusion seems to me to be formulated in a slightly too strong way.

I would formulate this argument ("LMs generally respond to animacy like humans") with more caution, emphasizing that it holds in the experiments performed and not as a general conclusion (after all, there could possibly be other experiments in which a different behavior might emerge, right?)

**Reproducibility:**

4: Could mostly reproduce the results, but there may be some variation because of sample variance or minor variations in their interpretation of the protocol or method.

**Reviewer Confidence:**

3: Pretty sure, but there's a chance I missed something. Although I have a good feel for this area in general, I did not carefully check the paper's details, e.g., the math, experimental design, or novelty.

**Typos Grammar Style And Presentation Improvements:**


You might want to choose colors that will also suit the color blind (or for printing in black and white)

line 104: "not only not only"

It would have been helpful if there had been more detail in the background about the tests themselves - what are they testing? when humans use them? what are their motivations? what was their purpose? and little clearer/detailed explanations in the introduction regarding what is atypical/typical animacy

---

> ### Author Rebuttal · Authors · 2023-08-28
>
> Thank you for your review! We’re happy to hear you found the paper enjoyable, interesting, and clear. We agree that the conclusion could be toned down—while LMs performed well on our experiments, these necessarily represent only a limited selection of the many ways in which animacy can manifest in language. We will revise the paper to make this nuance clearer.
>
> We’ll also take a closer look at the colors chosen for our graphs, and will consider adding symbols that make them more legible when printed in black and white. We will also add the requested details, with respect to e.g. a/typical animacy.

---

### Meta-Review · Area_Chair_iMuz · 2023-09-03

**Recommendation:** 5
**Confidence:** 5

**Metareview:**

This paper presents a study of animacy processing in LLMs, analogizing it to human processing as evidenced by psycholinguistic studies. The paper analyzes animacy processing in openly available datasets, such as BLIMP. However, the main contribution is adapting several Dutch psycholinguistic studies to evaluation by English LMs.

The reviewers unanimously gave very positive feedback, which I summarise in the following lines:

** Reasons To Accept: **

- The experiments are described clearly and contain relevant and interesting discussions.
- The paper further navigates its interdisciplinary setting very well by describing the psycholinguistic investigations and describing in enough detail its adaptation to LM evaluation.

** Reasons To Reject: **

- The background section is missing previous work on animacy detection.
- Section 5 as mentioned by the third reviewer, should be refined to have a very competitive paper.

---

### Decision · Program_Chairs · 2023-10-07

**Decision:**

Accept-Main

**Comment:**

This paper presents a study of animacy processing in LLMs, analogizing it to human processing as evidenced by psycholinguistic studies. The paper analyzes animacy processing in openly available datasets, such as BLIMP. However, the main contribution is adapting several Dutch psycholinguistic studies to evaluation by English LMs.

The reviewers unanimously gave very positive feedback, which I summarise in the following lines:

** Reasons To Accept: **

- The experiments are described clearly and contain relevant and interesting discussions.
- The paper further navigates its interdisciplinary setting very well by describing the psycholinguistic investigations and describing in enough detail its adaptation to LM evaluation.

** Reasons To Reject: **

- The background section is missing previous work on animacy detection.
- Section 5 as mentioned by the third reviewer, should be refined to have a very competitive paper.